# Feasibility of Stationary Cycling with Pedal Force Visual Feedback Post-Total Knee Arthroplasty: Implications for Inter-Limb Deficits in Knee Joint Biomechanics

**DOI:** 10.3390/bioengineering11080850

**Published:** 2024-08-20

**Authors:** Erik T. Hummer, Jared Porter, Harold Cates, Songning Zhang

**Affiliations:** 1Department of Kinesiology and Health, Rutgers, The State University of New Jersey, New Brunswick, NJ 08901, USA; 2Department of Physical Medicine and Rehabilitation, Rutgers New Jersey Medical School, Newark, NJ 07103, USA; 3Department of Kinesiology, Recreation and Sport Studies, University of Tennessee, Knoxville, TN 37996, USA; jporter@utk.edu (J.P.); szhang@utk.edu (S.Z.); 4Tennessee Orthopedic Clinics, Knoxville, TN 37923, USA; cateshe@tocdocs.com

**Keywords:** osteoarthritis, feedback, kinetics, gait analysis

## Abstract

The purpose of this study was to assess the biomechanical adaptations prompted by stationary cycling paired with visual feedback of vertical pedal reaction forces during both stationary cycling and overground walking for patients who underwent a total knee arthroplasty (TKA). Specifically, an emphasis on the inter-limb deficits in knee joint biomechanics were examined. Ten patients who underwent a TKA took part in an acute intervention with pre- and post-testing measurements of kinematics (240 Hz) and kinetics (1200 Hz) during stationary cycling and overground walking. The intervention phase consisted of six cycling sessions during which participants were provided with visual feedback of their bilateral peak vertical pedal reaction force, with instructions to maintain a symmetrical loading between limbs. A 2 × 2 (work rate/speed × time) repeated measures ANOVA (α = 0.05) was conducted on key outcome variables. Peak knee extension moment asymmetry during stationary cycling significantly improved (*p* = 0.038, η^2^_p_ = 0.610) following the acute intervention. Walking velocities for both preferred (*p* = 0.001, d = 0.583) and fast (*p* = 0.002, d = 0.613) walking speeds displayed improvements from pre- to post-testing. Significant improvements in the total score (*p* = 0.009, d = 0.492) and ADL subscale score (*p* = 0.041, d = 0.270) for the Knee Injury and Osteoarthritis Outcome Score were present following the acute intervention. Stationary cycling with visual feedback may be beneficial post-TKA; however, further investigation is merited.

## 1. Introduction

Total knee arthroplasty (TKA) is performed to restore knee joint function, alignment, and alleviate pain for individuals with end stage knee osteoarthritis; however, joint function may not be fully restored post-operation [1]. Disproportionate inter-limb loading deficits for the internal knee extension moment (KEM) often persist post-operation [1,2], increasing the risk for joint arthroplasties of the contralateral limb or revision of the primary implant [3]. Individuals post-TKA display inter-limb deficits in measures of quadriceps strength, with the operated limb exhibiting significantly decreased strength compared to the non-operated [4,5,6]. Prolonged deficits in quadriceps strength post-TKA raise additional concerns relating to functional abilities [7] and marked decreases in gait velocities that are present post-operation [8].

Dissimilar to patients diagnosed with knee osteoarthritis [9], no widely recognized rehabilitation guidelines exist for patients post-TKA. A common rehabilitation recommendation is to partake in activities eliciting lower tibiofemoral joint loading, such as stationary cycling [10,11]. In addition to experiencing lower knee joint loading, stationary cycling has been found to improve cardiovascular health, muscular strength, and muscular endurance [12]. However, significant inter-limb deficits of knee joint moments (e.g., knee extension moments) and pedal kinetics (e.g., vertical pedal reaction force) have been identified during stationary cycling [13] similar to those present in gait [5,14,15] post-TKA. Deficits in KEMs may be indicative of altered loading of the tibiofemoral joint [16] as well as quadriceps involvement and strength during activity. Concurrently, there is limited evidence supporting the efficacy of stationary cycling post-TKA in the literature [17]. When compared, a stationary cycling-focused rehabilitation program compared to a standard care program post-TKA found no significant benefits [17]. However, the intervention was introduced very early in the rehabilitation program, did not progress intensity, and outcome measures were limited to the Western Ontario and McMaster Universities Osteoarthritis Index (WOMAC). Understanding the impact of cycling rehabilitation on knee joint biomechanics may provide evidence-based recommendations for the use of stationary cycling post-TKA.

Augmented feedback (e.g., visual feedback) has been employed to enhance the benefits of cycling training paradigms in clinical populations [18,19,20]. Visual feedback stimuli range from using torque generated at the crank [20] to tangential pedal force [18,19]. Promising findings have been reported with improvements in gait speed for patients post-stroke compared to control groups [19]. While cycling with visual feedback can be beneficial, currently, there is limited research demonstrating feasibility of its use in reducing inter-limb loading deficits for patients post-TKA. Furthermore, if visual feedback could be used to specifically target measures of knee extensor involvement (KEM), then improvements in functionality and gait velocities may be possible utilizing stationary cycling.

The purpose of this study was to examine how stationary cycling paired with visual feedback of the bilateral peak vertical pedal reaction force impacts biomechanical inter-limb asymmetries in stationary cycling and gait post-TKA. Our primary hypothesis was that inter-limb asymmetries for the peak KEM, and the vertical and posterior pedal reaction forces, during cycling would improve pre- to post-training. Our secondary hypothesis was that the inter-limb asymmetries for peak KEM and vertical ground reaction force (GRF) during overground walking would improve pre- to post-training. Thirdly, we hypothesized that gait velocities, performance on functional tests, and Knee Injury and Osteoarthritis Outcome Scores (KOOSs) would improve post-intervention.

## 2. Materials and Methods

### 2.1. Participants

Ten participants were recruited from a local orthopedic clinic following unilateral TKAs performed by the same surgeon (Table 1). Participants were enrolled if they met the inclusion criteria while not satisfying any of the exclusion criteria (Table 2). All procedures were approved by the University Institutional Review Board and all participants completed an informed consent prior to participation.

### 2.2. Procedures

#### 2.2.1. Instrumentation

Three-dimensional kinematics were collected using a 12-camera motion capture system (240 Hz, Vicon Motion Capture Inc., Oxford, UK). Individual reflective markers and rigid clusters were used to track segmental motion of the trunk, pelvis, thighs, shanks, and feet [13,21]. Reflective markers were placed on each pedal and crank arm axis to track pedal motion [13,22]. Three-dimensional pedal reaction force data during cycling were collected using two instrumented pedals outfitted with two tri-axial force sensors (1200 Hz, Type 9027C, Kistler, Switzerland) and two amplifiers (Type 5073A, Kistler, Switzerland) [13,23]. Three-dimensional GRF data during gait were recorded with an imbedded force platform (1200 Hz, Advanced Mechanical Technology Inc., Watertown, MA, USA). Kinematic and kinetic data were collected simultaneously with the Nexus software suite (version 2.8.2, Vicon Motion Capture Inc., Oxford, UK).

#### 2.2.2. Testing Protocol

Testing included two identical sessions (pre- and post-training), separated by a 6-session intervention protocol. First, participants completed the KOOS questionnaire followed by changing into standard laboratory shoes (Pegasus 32, Nike Inc., Portland, OR, USA). Participants then completed a three-minute treadmill warm up at a self-selected speed followed by two trials of a ten-minute timed up and go test [24] and 10-repetition sit-to-stand test [25]. Next, participants underwent stationary cycling (Excalibur, Lode B.V., Groningen, The Netherlands) at two randomized work rates (80 and 100 W) and 80 revolutions per minute (RPM). The stationary ergometer was set for each participant based on saddle height [22,26], saddle fore-aft position [22], and handlebar position [23]. For cycling conditions, participants cycled for one minute, with data collection occurring in the final ten seconds.

After cycling, five level walking trials in four conditions consisting of two speeds (preferred and fast) and limb conditions (replaced and non-replaced), randomized respectively in that order, were conducted. Gait speeds were recorded using two photocells (63501 IR, Lafayette Instrument Inc., Lafayette, IN, USA) three meters apart. Preferred speed was at a velocity representative of participants’ typical walking, assessed over three practice trials. The fast speed condition was set at the participants’ preferred speed + 0.4 m/s. Trials were successful if participants were within ±10% of the desired speed. Upon arrival and with interceding gait trials, participants rated the perceived pain for their replaced knee using an enlarged VNS ranging from 0 to 10.

#### 2.2.3. Intervention Protocol

Participants attended six training sessions consisting of multiple 5 min cycling bouts over 3 weeks. The first session consisted of two bouts (5 min × 2 bouts = 10 min), the second session consisted of three bouts, and the remaining had four bouts. Bouts were interceded by a minimum one-minute rest, during which participants were asked to rate their replaced knee pain and their exertion using the Borg 6–20 rating of perceived exertion (RPE) scale [27]. Cadence was maintained throughout training at 80 RPM. Work rate was set between each bout of exercise, initially starting at 60 W. Work rate was moderated between bouts based on VNS pain, RPE, and the asymmetry index of peak vertical pedal reaction force (Equation (1), Figure 1).
(1)$$Asymmetry\;Index=\frac{X_{non-replaced}-X_{replaced}}{X_{non-replaced}}\times 100$$
where X_non-replaced_ is the peak variable for the non-replaced limb and X_replaced_ is the peak variable for the replaced limb. An asymmetry index of zero indicates a complete symmetry, whereas increasing asymmetry index values away from zero indicate larger asymmetries. These criteria were chosen to reduce asymmetries while maintaining low levels of pain and promoting a moderate intensity of exercise (RPE = 15).

Within bouts, participants were provided with terminal summary visual feedback of their peak vertical pedal reaction force for both limbs. Vertical pedal reaction force data were collected, processed, and displayed using MATLAB (2019a, MathWorks, Natick, MA, USA). Data were collected for a total of 30 s, starting at twenty seconds and every minute thereafter (e.g., 1 min 20 s). Visual feedback was displayed for a duration of twenty seconds. Raw vertical pedal reaction force data were filtered using a fourth-order zero lag Butterworth lowpass filter (6 Hz) [23]. The peak vertical pedal reaction force was identified for each crank cycle and an average was computed for both limbs.

Visual feedback was presented as bars corresponding to their right and left limbs (Figure 2). Participants were given a target range of ±10% of the average peak vertical pedal reaction force for their non-replaced limbs, represented as horizontal lines. Participants were instructed to keep both bars within this target range, intending to elicit inter-limb asymmetries of 10% or less [28]. During the first training session, participants were familiarized with how to interpret the feedback (Figure 2).

### 2.3. Data Analysis

Five individual trials for each condition were analyzed. Kinematics and kinetics expressed using the right-hand rule were calculated in Visual3D (Version 6.01, C-Motion Inc., Germantown, MD, USA). Marker trajectories and pedal reaction force data were filtered using a fourth-order zero lag Butterworth lowpass filter (6 Hz) [22,26]. Raw GRF data were filtered using a fourth-order zero lag Butterworth lowpass filter (50 Hz) [29]. Joint angular kinematics were calculated using the joint coordinate system and a Cardan rotational sequence (X-Y-Z) [30]. Joint moments during cycling and pedal reaction force data were not normalized to body mass [22]. Net joint moments during walking were normalized to body mass (Nm/kg) and GRF data were normalized to body weight (BW) [29].

Asymmetry indices for each peak variable: KEM, vertical pedal reaction force, posterior pedal reaction force, and vertical GRF, were computed (Equation (1)). Of the ten participants, seven were identified as ‘responders’ with notable improvements to their peak KEM asymmetry index during cycling from pre- to post-training. The other three participants either did not positively respond (*n* = 2) or had results that were statistical outliers (*n* =1) exceeding two standard deviations. Data of the responders were included for statistical analyses for all key variable outcomes.

### 2.4. Statistical Analysis

#### 2.4.1. Hypotheses 1 and 2

A 2 × 2 (work rate × time) repeated measures analysis of variance (ANOVA) was run on key outcome variable asymmetry indices comparing pre- and post-training measurements during stationary cycling and gait. Alpha levels were set *a priori* at 0.05. Effect sizes were reported as partial eta squared (η^2^_p_) and were interpreted as small (η^2^_p_ < 0.06), medium (0.06 ≤ η^2^_p_ < 0.15), and large (η^2^_p_ ≥ 0.15) [31].

#### 2.4.2. Hypothesis 3

Paired *t*-tests (α = 0.05) were run to compare gait velocities, VNS pain scores, functional test outcomes, and KOOSs between pre- and post-training, accompanied with Cohen’s D effect sizes interpreted as small (d ≤ 0.20), medium (0.21 < d ≤ 0.50), and large (0.51 < d ≤ 0.80) [32].

## 3. Results

### 3.1. Hypothesis 1: Cycling Asymmetries

The peak KEM asymmetry index displayed a significant effect of time (*p* = 0.038, η^2^_p_ = 0.610) with improvements in pre- to post-training [mean difference (MD) = −30.3%, Table 3]. There was a significant interaction for peak vertical pedal reaction force asymmetry index (*p* = 0.032, η^2^_p_ = 0.634) with larger changes to asymmetries at 80 W (15.0%) compared to 100 W (6.7%). There was a large non-significant effect of time for the peak posterior pedal reaction force asymmetry index (*p* = 0.057, η^2^_p_ = 0.549), displaying an improvement post-training (MD = −18.4%).

### 3.2. Hypothesis 2: Overground Walking Asymmetries

Large non-significant effects of time for the peak KEM asymmetry index during load-response (*p* = 0.382, η^2^_p_ = 0.194, MD = −16.3%) and push-off (*p* = 0.134, η^2^_p_ = 0.468, MD = −20.1%) (Table 4). Push-off KEM asymmetry displayed a significant improvement (*p* = 0.031, η^2^_p_ = 0.726) between preferred and fast speeds (MD = −16.6%).

### 3.3. Hypothesis 3: Gait Velocities, Functional Tests, and KOOS

There were significant increases between pre- and post-training gait velocities for preferred (*p* = 0.001, d = 0.583) and fast (*p* = 0.001, d = 0.613) speeds (Table 5). No significant differences were found for the timed up and go (*p* = 0.232, d = 0.315) or the sit-to-stand test (*p* = 0.807, d = 0.059). There was a significant improvement in the total KOOS (*p* = 0.009, d = 0.492) and the activities of daily living subscale (*p* = 0.041, d = 0.270).

## 4. Discussion

The purpose of this study was to examine how stationary cycling paired with visual feedback of peak vertical pedal reaction force impacts biomechanical inter-limb asymmetries during stationary cycling and overground walking post-TKA.

Our primary hypothesis was partially supported, with an observed significant improvement (MD = −30.3%) in the peak KEM asymmetry index during cycling from pre- to post-training. Typically, a threshold of a 10% improvement in the asymmetry index is considered clinically relevant [28], suggesting that stationary cycling with visual feedback of the peak vertical pedal reaction force could be effective in improving KEM inter-limb asymmetries in patients post-TKA. However, these beneficial adaptations were not observed when all three key outcome variables (KEM, vertical pedal reaction force, and posterior pedal reaction force) were aggregated within multivariate analysis. The significant improvement of KEM asymmetry may be the result of combined changes observed in the peak vertical and posterior pedal reaction force asymmetry index, both of which showed large effect sizes (η^2^_p_ = 0.168 and 0.549, respectively). Interestingly, despite visual feedback being presented of the peak vertical pedal reaction force, there was a much larger change in peak posterior pedal reaction force asymmetries (MD = 18.4%). This larger training effect in peak posterior pedal reaction force asymmetry could be due to its larger asymmetry index at pre-training compared to the vertical pedal reaction force asymmetry index (27.6% vs. 2.5%). Further exploratory correlation analysis (Spearman rank) found that the correlations for pre-training asymmetries between the posterior pedal reaction force and KEM (*p* = 0.895, *p* < 0.001) were stronger than those between the vertical pedal reaction force and KEM (*p* = 0.476, *p* = 0.188). Utilizing the sagittal plane pedal reaction force (resultant of vertical and anterior-posterior pedal reaction force components) may be more beneficial in targeting improvements in cycling-related asymmetries in knee joint biomechanics and merits further investigation [33].

Our secondary hypothesis was not supported, with no significant training effects for the overground walking asymmetries in KEM or vertical GRF (Table 4). Despite the lack of statistical significance, peak KEM asymmetry during load-response and push-off did display large effect sizes (η^2^_p_ = 0.194 and 0.468, respectively) for time. Improvements in KEM asymmetries for both of these critical points during the gait cycle ranged from 16.3 to 20.1%, which may indicate a clinically relevant shift to symmetrical knee joint loading between the replaced and non-replaced limbs [28]. Achieving a symmetrical loading can be crucial, as exacerbated inter-limb loading deficits have been a cause for concern for revision of the primary TKA or TKA of the contralateral limb [3]. The large changes in KEM asymmetry were not accompanied by equivalent changes in vertical GRF asymmetries (−3.6–1.7%), which were more symmetrical at pre-training. One potential explanation could be due to the progressive intensity of the cycling intervention. High-intensity training has been shown to elicit positive effects for both patients with TKA and knee OA [34,35]. These results may indicate that, given a larger dose of training or with more participants, there may be an underlying transfer effect to gait-related asymmetries that merits further investigation.

Our tertiary hypothesis was partially supported, with significant improvements for preferred and fast gait velocities and the ‘activities of daily living’ KOOS subscale following the training (Table 5). Previous cycling interventions have been found to elicit increases in gait velocity for other clinical populations, such as cerebral palsy and stroke patients [19,36]. One factor that was postulated to impact potential benefits is the severity of impairment. Those participants with a greater gait impairment post-hemiplegic stroke [19] and trained cyclists with greater asymmetries [37] displayed a greater benefit from cycling interventions. Our participants were 8.6 months post-operation on average and not currently experiencing significant pain in their replaced limbs (VNS = 0.6). Perhaps introducing stationary cycling with visual feedback much earlier in their rehabilitation, when patients may be experiencing greater impairments, would elicit greater adaptation. Despite improvements in the overground walking speed, there was no significant improvement (+6.2%) in the timed up and go test, which was accompanied with a medium effect size (d = 0.385). Finally, the responders did show significant improvements in the ‘activities of daily living’ KOOS subscale, but only with a medium effect. Cycling with visual feedback of the peak vertical pedal reaction force may prove to be beneficial in improving walking speeds and other assessments of functionality, and merits further investigation.

The current study is not without limitations. First, due to difficulties with recruitment and testing due to the COVID-19 pandemic, we were unable to fully collect the remaining participants and finish the intervention of two participants to reach the desired statistical power. The smaller sample size may contribute to the lack of significant changes in some findings and may limit the conclusions of the current findings. However, medium to large effect sizes did provide some support for our findings. Another limitation was a lack of control group. Similarly, this was due to difficulties imposed by the COVID-19 pandemic. This comparison would aid in examining if changes seen in the intervention group were due to the intervention, or potentially due to time and recovery. Third, the current intervention included only six training sessions, potentially limiting the dosing of the intervention. A greater dosage of the intervention (e.g., more training sessions over a longer period) may provide information on the long-term impacts of this training program. Fourth, we provided feedback on the peak vertical pedal reaction force to elicit changes in the peak KEM during cycling. Our participants did not show significant asymmetry in vertical pedal reaction force in their replaced limbs prior to training, but they did show large asymmetries in the peak KEM and posterior pedal reaction force. Utilizing either the KEM directly or the sagittal plane pedal reaction force (vertical and anterior/posterior components) may provide more promising adaptations to training.

## 5. Conclusions

The current cycling intervention paired with visual feedback significantly altered peak KEM asymmetry during cycling and improved overground walking speeds. These findings may indicate a clinically relevant improvement in KEM asymmetry and walking capacity. However, there were no observed adaptations in KEM or vertical GRF asymmetries during overground walking, albeit large effect sizes were present. Further work is needed to fully explore the benefits of stationary cycling with visual feedback, including a larger participant pool, increased dosage of the intervention (e.g., more training sessions), and/or implementation of the intervention earlier in rehabilitation closer to post-operation. Finally, investigating the optimal visual feedback cue remains a high priority.

## Figures and Tables

**Figure 1 bioengineering-11-00850-f001:**
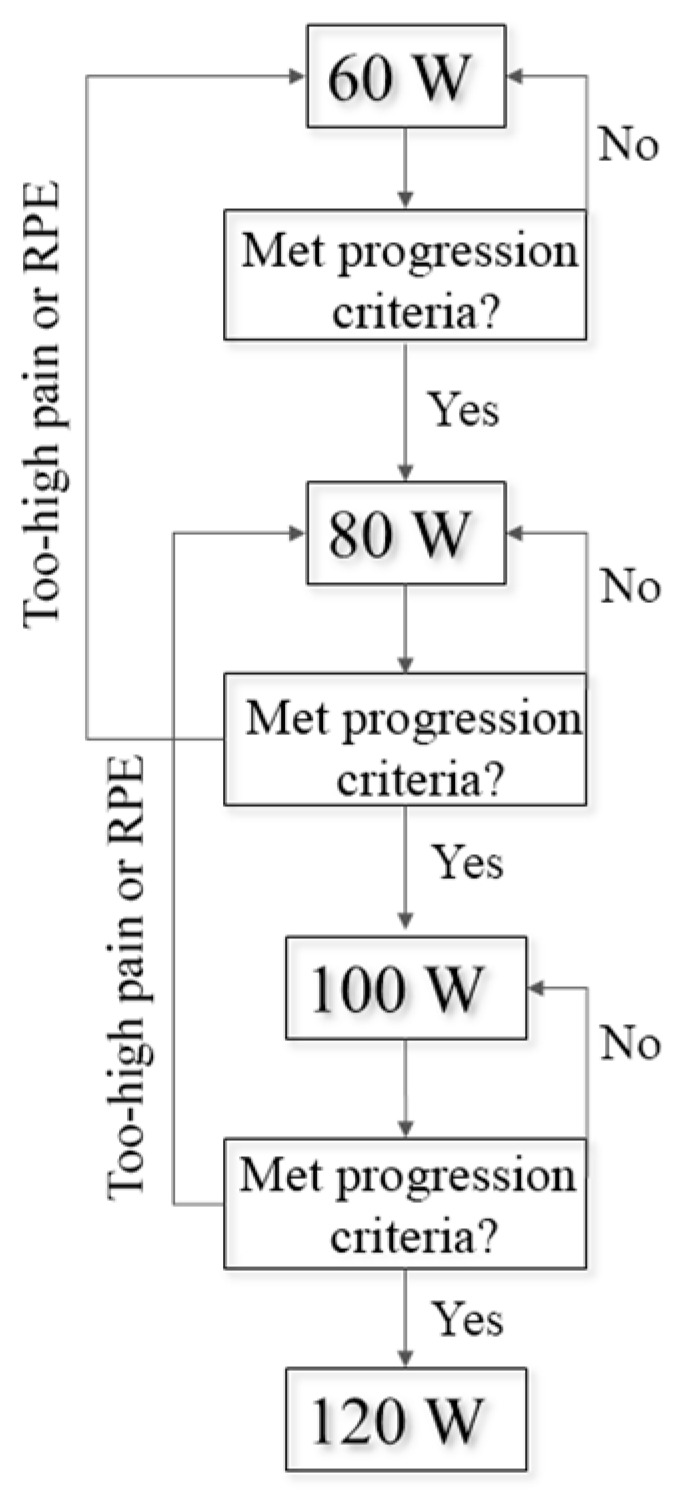
The intervention progression plan. Work rates were increased in 20 W increments if the visual numeric scale (VNS) for replaced knee pain was <2 from the previous level, if rating of perceived exertion was <15, and if their pedal asymmetry index was <20%. Work rate was maintained if VNS replaced knee pain was <2 from the previous recording, RPE = 15, or if pedal AI was >20%. Work rate was reduced by 20 watts if VNS replaced knee pain was ≥+2 from the previous recording or if RPE was >15.

**Figure 2 bioengineering-11-00850-f002:**
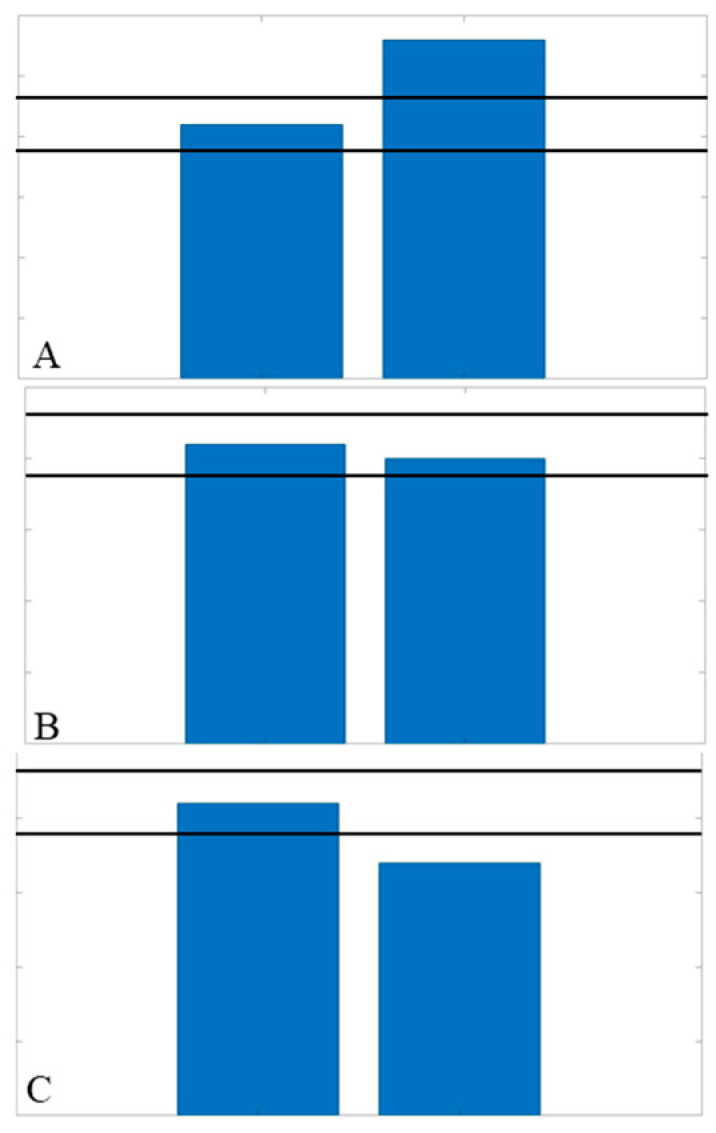
Examples of potential feedback displays shown to each participant on the first day of training. The right and left bars correspond to the right and left limb, respectively. (**A**) Display of greater force on the right limb, exceeding the threshold (horizontal black bars indicating ±10% of the non-replaced force). (**B**) Display of both sides within the target threshold, indicating the ideal symmetrical condition. (**C**) Display of the right limb having a decreased force, indicating being below the target threshold.

**Table 1 bioengineering-11-00850-t001:** Participant demographic characteristics including age (years), mass (kg), height (m), and time post-operation (months) (mean ± STD) and the inclusion/exclusion criteria used during recruitment.

	Characteristics
Age	64.8 ± 7.7
Mass	89.2 ± 21.3
Height	1.70 ± 0.1
Time post-operation	8.6 ± 2.4

**Table 2 bioengineering-11-00850-t002:** Inclusion and exclusion criteria utilized to screen potential participants prior to recruitment.

Inclusion	Exclusion
◾ Age: 50–80 years ◾ Time post-operation: 6–18 months	◾ Debilitating osteoarthritis of other lower extremity joints impacting their gait ◾ Arthroplasty of other lower extremity joints ◾ BMI ≥ 38 kg/m^2^ ◾ Neurological diseases that impact gait or balance ◾ Systemic inflammatory arthritis ◾ Arthroscopic surgery or lower limb injury within 6 months ◾ >5 level of pain in the replaced knee using a 0–10 VNS scale ◾ Required the use of walking aids

**Table 3 bioengineering-11-00850-t003:** Responders’ asymmetry index (%) for peak knee extension moment (KEM), vertical pedal reaction force (PRF), and posterior PRF during stationary cycling in pre- and post-training sessions for 80 and 100 watts (mean ± STD).

	80 W		100 W			P (η^2^_p_)	
	Pre	Post	Pre	Post	Interaction	Time	Work Rate
KEM	36.0 ± 22.7	−2.0 ± 14.4	33.2 ± 17.7	10.6 ± 11.4	0.109 (0.432)	**0.038** (0.610)	0.262 (0.242)
Vertical PRF	3.6 ± 7.7	−11.5 ± 21.8	1.2 ± 14.5	−5.3 ± 15.3	**0.032** (0.634)	0.362 (0.168)	0.548 (0.077)
Posterior PRF	26.2 ± 15.5	6.5 ± 3.6	29.0 ± 15.9	11.9 ± 8.3	0.537 (0.081)	0.057 (0.549)	**0.050** (0.570)

KEM—knee extension moment; η^2^_p_—partial eta squared; bolded values indicate significance (*p* < 0.05).

**Table 4 bioengineering-11-00850-t004:** Responders’ asymmetry index (%) for peak knee extension moment (KEM) and vertical ground reaction force (GRF) AI during overground gait for preferred and fast walking speeds in pre- and post-training (mean ± STD).

	Preferred		Fast		P (η^2^_p_)		
	Pre	Post	Pre	Post	Interaction	Time	Speed
LR KEM	34.4 ± 36.6	14.8 ± 12.7	40.0 ± 16.9	26.9 ± 15.4	0.525 (0.108)	0.382 (0.194)	0.198 (0.373)
PO KEM	4.0 ± 17.4	−17.6 ± 27.0	22.1 ± 10.6	−2.5 ± 28.4	0.854 (0.009)	0.134 (0.468)	**0.031** (0.726)
LR vertical GRF	3.1 ± 2.8	3.7 ± 3.4	5.2 ± 3.3	3.4 ± 4.7	0.225 (0.339)	0.669 (0.050)	0.182 (0.395)
PO vertical GRF	3.4 ± 1.4	5.1 ± 1.2	7.0 ± 3.6	3.4 ± 7.1	0.080 (0.575)	0.351 (0.218)	0.611 (0.071)

LR—load-response; KEM—knee extension moment; PO—push-off; GRF—ground reaction force; η^2^_p_—partial eta squared effect size; bolded values indicate significance (*p* < 0.05).

**Table 5 bioengineering-11-00850-t005:** Gait speeds, VNS pain scores, functional tests’ outcomes, and KOOSs for the responder participants in pre- and post-training (mean ± STD).

	Pre-Training	Post-Training	*p*	d
Gait Speeds				
Preferred Gait Velocity (m/s)	1.21 ± 0.23	1.35 ± 0.25	**0.001**	0.583
Fast Gait Velocity (m/s)	1.54 ± 0.18	1.67 ± 0.24	**0.002**	0.613
VNS Pain Score				
Initial	0.60 ± 1.34	0.95 ± 1.45	0.343	0.251
Preferred Speed	0.60 ± 1.58	0.60 ± 1.26	1.000	0.000
Fast Speed	0.65 ± 1.56	0.60 ± 1.26	0.758	0.035
Functional Tests				
Timed-up-and-go (s)	8.49 ± 1.65	7.96 ± 1.71	0.232	0.315
Sit-to-Stand (s)	24.58 ± 5.72	24.19 ± 7.38	0.807	0.059
KOOS				
Total Score	339.2 ± 50.4	361.5 ± 39.9	**0.009**	0.492
Symptom’s subscale	79.76 ± 17.00	83.93 ± 12.93	0.272	0.276
Pain subscale	87.96 ± 14.66	93.23 ± 9.70	0.105	0.424
ADL subscale	94.36 ± 5.13	95.83 ± 5.76	**0.041**	0.270
Quality of Life subscale	77.08 ± 19.63	88.54 ± 16.02	0.100	0.640

VNS—Visual numeric scale; initial—start of the testing session; KOOS—Knee Injury and Osteoarthritis Score; ADL—activities of daily living; d—Cohen’s D effect size; bolded values indicate significance (*p* < 0.05).

## Data Availability

The raw data supporting the conclusions of this article will be made available by the authors on request.

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
