# Peer review of "Feasibility of Stationary Cycling with Pedal Force Visual Feedback Post-Total Knee Arthroplasty: Implications for Inter-Limb Deficits in Knee Joint Biomechanics"

_bioengineering, 2024, doi:10.3390/bioengineering11080850_

Round 1

Reviewer 1 Report

Comments and Suggestions for Authors

1.     The introduction section needs substantial enhancement. To ensure a systematic understanding of the topic, the authors should elaborate more comprehensively. They must include robust literature to develop and clearly convey the central idea of the article.

2.     Please update the references where DOI links are missing, such as in references 2, 8, 24, and 27.

3.     What are the primary biomechanical deficits observed in patients after total knee arthroplasty?

4.     How does inter-limb asymmetry in knee joint biomechanics affect long-term recovery and function post-TKA?

5.     What are the key parameters to measure and monitor during stationary cycling to assess knee joint biomechanics?

6.     How do different cycling protocols (e.g., duration, intensity, frequency) affect rehabilitation outcomes in TKA patients?

7.     What is the role of muscle strength and coordination in addressing inter-limb deficits post-TKA during stationary cycling?.

8.     What are the biomechanical and functional performance improvements seen in patients who use pedal force visual feedback compared to those who do not?

Comments on the Quality of English Language

The manuscript needs to be thoroughly checked for grammatical errors and typos.

Author Response

  1. The introduction section needs substantial enhancement. To ensure a systematic understanding of the topic, the authors should elaborate more comprehensively. They must include robust literature to develop and clearly convey the central idea of the article.

We greatly appreciate your time and effort reviewing our manuscript. We have undergone a revision of our introduction to expand more on the topics you have recommended to appropriately convey and support the central idea of our manuscript. We believe that our manuscript has benefited greatly from your comments. The revised text in our manuscript will be reflected as red font in addition to responses to your specific comments below.

  1. Please update the references where DOI links are missing, such as in references 2, 8, 24, and 27.

We have revised our manuscript to include an updated reference list to include the DOI links for references that were missing in our original submission.

3. What are the primary biomechanical deficits observed in patients after total knee arthroplasty?

We have revised the manuscript on lines 33-38 to include more discussion on the specific biomechanical deficits observed post total knee arthroplasty. These include deficits in quadriceps strength, knee extension moment (both during gait and stationary cycling), and measures of reaction forces (both ground reaction force during gait and pedal reaction force during cycling).

4. How does inter-limb asymmetry in knee joint biomechanics affect long-term recovery and function post-TKA?

We have revised the manuscript on lines 38-40 to include more pertinent information on the long-term recovery post total knee arthroplasty. Specifically, we have included more detailed information on how deficits in lower extremity strength are related to functional abilities for patients’ post-operation as well as decreases in gait velocities, which is a commonly used to assess overall health and fall risk for an aging population. Finally, we have included more details displaying that altered loading can lead to an increased risk of revision of the primary replacement or secondary replacement of the non-operated limb.

5. What are the key parameters to measure and monitor during stationary cycling to assess knee joint biomechanics?

We have revised the manuscript on lines 46-50 to include more specific details expanding on the important role of knee extension moments as it pertains to both tibiofemoral loading as well as involvement of the quadriceps muscles during activities. These are critical as altered loading of the tibiofemoral joint may be cause for concern relating to revision of the primary replacement or operation on the contralateral limb. Additionally, using knee extension moment as an assessment of quadriceps involvement is critical to allow individuals to target strength deficits that related directly to functional abilities and gait velocities [1,2].

6. How do different cycling protocols (e.g., duration, intensity, frequency) affect rehabilitation outcomes in TKA patients?

Unfortunately, there is still limited research available for cycling specific protocols for patients following total knee arthroplasty. Two studies did include the use of stationary cycling in rehabilitation programs [3,4] with durations ranging from 5 to 20 minutes per session over the course of 6 or eight weeks. Since the programs included in the two cited studies utilized other rehabilitation modalities (e.g. lower extremity strengthening and neuromuscular training), it may not be directly relatable to the current study only utilizing stationary cycling.

  1. What is the role of muscle strength and coordination in addressing inter-limb deficits post-TKA during stationary cycling?

      We have revised the manuscript on lines 49-50 and 65-67 to include a more detailed exploration on the role of quadriceps strength and its relationship to recovery from a total knee arthroplasty. We have also included a deeper exploration on addressing muscle strength using knee extension moment during cycling.

  1. What are the biomechanical and functional performance improvements seen in patients who use pedal force visual feedback compared to those who do not?

      Unfortunately, we are currently limited when attempting to compare adaptations in knee joint biomechanics between those receiving visual feedback compared to those who are not. One study examined adaptations in the Knee Injury and Osteoarthritis Outcome Score (KOOS) following a cycling rehabilitation program [5]. Our original intention from this work was to include a control group receiving the intervention without visual feedback, however due to the COVID-19 pandemic we were unable to recruit participants for the control group prior to the PhD student’s graduation. We acknowledge that this merits further investigation.  

1. Mizner, R.L.; Petterson, S.C.; Stevens, J.E.; Axe, M.J.; Snyder-Mackler, L. Preoperative quadriceps strength predicts functional ability one year after total knee arthroplasty. J Rheumatol 2005, 32, 1533-1539.

2. Mizner, R.L.; Petterson, S.C.; Stevens, J.E.; Vandenborne, K.; Snyder-Mackler, L. Early quadriceps strength loss after total knee arthroplasty. The contributions of muscle atrophy and failure of voluntary muscle activation. J Bone Joint Surg Am 2005, 87, 1047-1053, doi:10.2106/JBJS.D.01992.

3. Stevens, J.E.; Mizner, R.L.; Snyder-Mackler, L. Quadriceps strength and volitional activation before and after total knee arthroplasty for osteoarthritis. J Orthop Res 2003, 21, 775-779, doi:10.1016/S0736-0266(03)00052-4.

4. Moffet, H.; Collet, J.P.; Shapiro, S.H.; Paradis, G.; Marquis, F.; Roy, L. Effectiveness of intensive rehabilitation on functional ability and quality of life after first total knee arthroplasty: A single-blind randomized controlled trial. Arch Phys Med Rehabil 2004, 85, 546-556, doi:10.1016/j.apmr.2003.08.080.

Reviewer 2 Report

Comments and Suggestions for Authors

The authors investigated the effects of stationary cycling paired with visual feedback of bilateral peak vertical pedal reaction force on biomechanical interlimb asymmetries during stationary cycling and gait post total knee arthroplasty. It was found that cycling paired with visual feedback significantly improved peak KEM asymmetry and preferred and fast walking speeds. Additionally, significant improvements were found in the total and ADL subscale scores for Knee Injury and Osteoarthritis Outcome Score. This study was carefully conducted, and the manuscript was well written. The following are comments for improving the manuscript.  

1.       There are some grammatical errors in the abstract.

2.       Had the authors or previous studies conducted a rehabilitation study using cycling paired without visual feedback of pedal reaction force for comparison?  

Comments on the Quality of English Language

Some grammatical errors need to be corrected. 

Author Response

The authors investigated the effects of stationary cycling paired with visual feedback of bilateral peak vertical pedal reaction force on biomechanical interlimb asymmetries during stationary cycling and gait post total knee arthroplasty. It was found that cycling paired with visual feedback significantly improved peak KEM asymmetry and preferred and fast walking speeds. Additionally, significant improvements were found in the total and ADL subscale scores for Knee Injury and Osteoarthritis Outcome Score. This study was carefully conducted, and the manuscript was well written. The following are comments for improving the manuscript.  

We greatly appreciate your time and effort in reviewing our manuscript. We have revised our manuscript according to your recommendations. The revised text can be found in the manuscript file in red font, as well as listed below with each specific point raised.

  1. There are some grammatical errors in the abstract.

We greatly appreciate you bringing these errors to our attention. We have revised the abstract considerably to address the grammatical errors that were present in our original submission. The revisions can be found in our revised manuscript file in red font color.

  1. Had the authors or previous studies conducted a rehabilitation study using cycling paired without visual feedback of pedal reaction force for comparison?  

Upon extensive literature review, we have only found one study that examined the impact of cycling for rehabilitation following total knee arthroplasty without visual feedback [5]. Unfortunately, that study only examined outcomes related to the Knee Injury and Osteoarthritis Outcome Score (KOOS). We have not collected data on cycling without visual feedback for this patient population. In our original design of this study, we aimed to collect data for a “control” group that would undergo cycling without visual feedback, however due to difficulties with the COVID-19 pandemic we were unable to collect that data prior to graduation for the PhD student conducting this study.

5. Liebs, T.R.; Herzberg, W.; Ruther, W.; Haasters, J.; Russlies, M.; Hassenpflug, J. Ergometer cycling after hip or knee replacement surgery: a randomized controlled trial. J Bone Joint Surg Am 2010, 92, 814-822, doi:10.2106/JBJS.H.01359.

Round 2

Reviewer 1 Report

Comments and Suggestions for Authors

The revisions are satisfactory now.

Comments on the Quality of English Language

Minor grammatical errors and typos to be corrected throughout the manuscript.